# Near-Infrared Autofluorescence (NIRAF) in Atherosclerotic Plaque Dissociates from Intraplaque Hemorrhage and Bilirubin

**DOI:** 10.3390/ijms241310727

**Published:** 2023-06-27

**Authors:** Weiyu Chen, James Nadel, Sergey Tumanov, Roland Stocker

**Affiliations:** 1Heart Research Institute, Sydney, NSW 2042, Australia; weiyu.chen@hri.org.au (W.C.); james.nadel@hri.org.au (J.N.);; 2Faculty of Medicine and Health, The University of Sydney, Sydney, NSW 2006, Australia; 3St Vincent’s Hospital, Sydney, NSW 2010, Australia; 4School of Clinical Medicine, University of New South Wales, Sydney, NSW 2052, Australia; 5School of Life and Environmental Sciences, The University of Sydney, Sydney, NSW 2006, Australia

**Keywords:** bilirubin, intraplaque hemorrhage (IPH), near-infrared autofluorescence (NIRAF), atherosclerosis

## Abstract

Near-infrared autofluorescence (NIRAF) in unstable atherosclerotic plaque has been suggested as a novel imaging technology for high-risk atherosclerosis. Intraplaque hemorrhage (IPH) and bilirubin, derived from the subsequent degradation of heme, have been proposed as the source of NIRAF, although their roles and the underlying mechanism responsible for NIRAF remain unclear. To test the proposed role of bilirubin as the source of NIRAF in high-risk atherosclerosis, *Biliverdin reductase a* gene and *apolipoprotein E* gene double-knockout (*Bvra^−/−^Apoe^−/−^*) mice were subjected to the Western diet and tandem stenosis (TS) surgery, as a model of both bilirubin deficiency and plaque instability. Human coronary arteries containing atherosclerotic plaques were obtained from heart transplant recipients. The NIRAF was determined by in vivo fluorescence emission computed tomography, and ex vivo infrared imaging. The cholesterol content was quantified by HPLC with UV detection. In *Bvra^+/+^Apoe^−/−^* TS mice, the NIRAF intensity was significantly higher in unstable plaque than in stable plaque, yet the NIRAF in unstable plaque was undistinguishable in *Bvra^+/+^Apoe^−/−^* and littermate *Bvra^−/−^Apoe^−/−^* TS mice. Moreover, the unstable plaque in TS mice exhibited a lower NIRAF compared with highly cellular plaque that lacked most of the features of unstable plaque. In human coronary arteries, the NIRAF associated with cholesterol-rich, calcified lesions, rather than just cholesterol-rich lesions. The NIRAF in atherosclerotic plaque can be dissociated from IPH and bilirubin, such that the compositional meaning of an elevated NIRAF remains obscure.

## 1. Introduction

Cardiovascular disease is the leading cause of death in the world. Its major cause is atherosclerosis, a progressive disease of large arteries that often culminates in the rupture of unstable atherosclerotic plaque. This can lead to thrombus formation, arterial occlusion, and acute cardiovascular events, e.g., heart attack or stroke. However, despite the potential life-threatening sequela, atherosclerotic plaques on the brink of rupture cannot be identified readily, as there is currently a lack of specific clinical imaging approaches that reliably target high-risk plaques.

The development of novel technologies to identify high-risk plaques has focused on their compositional and molecular features, including intraplaque hemorrhage (IPH). IPH leads to the induction of heme oxygenase 1, an enzyme that degrades heme to biliverdin, and the expression of which has been suggested to determine atherosclerotic lesion progression to a vulnerable plaque [1]. Recently, IPH was reported to be associated with an increased near-infrared autofluorescence (NIRAF) in the unstable plaque of both human endarterectomy specimens, and the right carotid artery of *apolipoprotein E* gene knockout (*Apoe*^−/−^) mice subjected to tandem stenosis (TS) [2]. Studies involving intravascular NIRAF imaging have also been conducted in human cadaver coronary arteries, to assess its feasibility and diagnostic value in patients with coronary artery disease [3,4]. These studies provide valuable insights into the application of NIRAF in a clinical setting. However, the prognostic impact of NIRAF on plaque progression remains unclear, and further insights into NIRAF generation are warranted.

Htun et al. reported bilirubin (formed from biliverdin by biliverdin reductase) to be the likely source of NIRAF, based on the co-localization of bilirubin with IPH in plaque, and the positive association between plaque bilirubin concentrations and NIRAF in the TS mouse model of plaque instability [2]. In apparent contrast, Albaghdadi et al. observed elevated NIRAF in the absence of co-localized IPH and bilirubin in human carotid endarterectomy specimens, and instead suggested an insoluble oxidized lipid or ceroid as a source of NIRAF [5]. Thus, the roles of IPH and bilirubin in the NIRAF of atherosclerotic plaque remain unclear; however, knowledge of the underlying molecular mechanism(s) is important to better understand the biological relevance and clinical significance of NIRAF-based imaging strategies.

To directly test the proposed role of bilirubin as the source of NIRAF, we crossed bilirubin-deficient *biliverdin reductase a* gene knockout (*Bvra*^−/−^) [6] with *apolipoprotein E gene* knockout (*Apoe*^−/−^) mice. We subjected the resulting *Bvra*^−/−^*Apoe*^−/−^ and littermate *Bvra*^+/+^*Apoe*^−/−^ control mice to tandem stenosis (TS) surgery [7], determined the in vivo NIRAF in the carotid arteries, and compared the NIRAF in isolated plaque-free and diseased arteries from these animals, representing different types of atherosclerotic plaques. We also determined the NIRAF in isolated human coronary arteries, and correlated the NIRAF with plaque constituents. The results obtained show that the NIRAF of atherosclerotic plaque can be dissociated from its bilirubin content, and that an elevated NIRAF appears to associate with mature complex plaque enriched in both cholesterol and calcification, rather than solely cholesterol-rich or IPH-containing lesions.

## 2. Results

### 2.1. Bilirubin Does Not Support a Broad Range of NIRAF in Atherosclerotic Plaque Containing Hemorrhage

We reported recently that the global deletion of the *biliverdin reductase a* (*Bvra*) gene renders *Apoe*^−/−^ TS mice deficient in bilirubin, and increases both plaque instability and IPH [8]. We therefore tested the role of bilirubin as the source of NIRAF, by subjecting *Bvra^+/+^Apoe^−/−^* and *Bvra^−/−^Apoe^−/−^* TS mice to non-invasive fluorescence emission computed tomography (FLECT/CT), using both the previously reported [2], and current, optimized settings. Contrary to a previous report [2], NIRAF was not observed in ten separate *Bvra^+/+^Apoe^−/−^* TS mice using in vivo FLECT/CT (Figure 1A). Similarly, in vivo NIRAF was also not seen in seven *Bvra^−/−^Apoe^−/−^* TS mice (Figure 1B). Despite this lack of discernible in vivo NIRAF in these animals, arterial segments representing unstable plaque contained IPH in both *Bvra^+/+^Apoe^−/−^* and *Bvra^−/−^Apoe^−/−^* TS mice, as observed by visual examination (see arrows in Figure 1A,B). Moreover, positive controls using Cy7 fluoroprobes in agarose gel, embedded in naïve mouse carcasses, showed an obvious NIRAF signal (Figure 1C), indicating the ability of FLECT/CT to detect NIRAF in vivo.

We then excised different arterial segments (Figure 2), representing different types of plaques, from *Bvra^+/+^Apoe^−/−^* and *Bvra^−/−^Apoe^−/−^* TS mice, and subjected the isolated segments to bright-field and infrared imaging (Figure 3A). The unstable plaque (Segment I) of *Bvra^+/+^Apoe^−/−^* TS mice had a significantly higher NIRAF than the stable plaque (Segment IV) and the lesion-free arteries (Segment V) (Figure 3B). However, the extent of the increased NIRAF signal in the unstable plaque (Segment I) of *Bvra^+/+^Apoe^−/−^* TS mice was comparable to that in *Bvra^−/−^Apoe^−/−^* TS mice (Figure 3B) that did not contain bilirubin [8]. Moreover, and irrespective of the presence of bilirubin/BVRA, the NIRAF signals were significantly lower among Segment I than Segment II, the latter containing highly cellular plaque, with features of stable plaque (Figure 3B). While these results confirm earlier studies reporting increased NIRAF in the IPH-containing unstable plaque of *Apoe^−/−^* TS mice [2], they also establish that bilirubin, and hence heme catabolism, is not the underlying source of the elevated NIRAF in unstable plaque.

### 2.2. NIRAF Reflects the Presence of Mature Atherosclerotic Disease

To better understand the molecular processes underlying elevated NIRAF in atherosclerotic plaque, we next determined NIRAF ex vivo, using segments of plaque-containing human coronary arteries. Thirty different coronary plaques were subjected to NIRAF (Figure 4A,B), ex vivo coronary artery computed tomography for plaque characterization [10], and total cholesterol determination using high-performance liquid chromatography, after the hydrolysis of tissue samples. The coronary segments analyzed contained mature (American Heart Association [AHA] grade IV–VI) atherosclerotic plaques, both with and without extensive calcification. Overall, a weak correlation between the coronary plaque cholesterol and the mean NIRAF was observed (Figure 4C), indicating that the accumulation of cholesterol alone is insufficient to significantly increase NIRAF. However, separating the coronary segments into calcific versus non-calcific plaques revealed a significant positive correlation between the cholesterol in the calcified plaques, and the mean NIRAF (Figure 4D). This result indicates that an elevation in the NIRAF in human coronary arteries may reflect the presence of a more complex, mature atherosclerotic disease.

## 3. Discussion

The present study, using a bilirubin-deficient mouse model of plaque instability, confirms that, compared with stable plaque, NIRAF is significantly increased in IPH-containing unstable plaque. What is new, however, is that neither bilirubin nor IPH is the likely source of NIRAF. In human coronary arteries, we also demonstrate that NIRAF is not solely associated with either cholesterol or calcification, but reflects the complexity and maturity of atherosclerotic plaque. Thus, the compositional meaning of an elevated NIRAF in atherosclerotic plaque remains obscure.

Our results confirm earlier studies that reported an increased NIRAF in the IPH-containing unstable plaque of *Apoe^−/−^* TS mice [2]. Intriguingly, there was no material difference in such NIRAF between the *Bvra^+/+^Apoe^−/−^* and *Bvra^−/−^Apoe^−/−^* TS mice. For reason(s) not clear at present, we were unable to detect NIRAF in unstable plaque using FLECT/CT, despite imaging a total of 17 TS mice, most of which contained visible hemorrhage in the arterial segment, where unstable plaque forms in this model of plaque instability. Notwithstanding this, our findings established that bilirubin, and hence heme catabolism, is not the underlying source of the elevated NIRAF in unstable plaque using, ex vivo, the NIRAF imaging system on the excised different arterial segments. This finding is contrary to the conclusion drawn by Htun et al., based on indirect evidence (the co-localization of bilirubin with IPH in plaque, and positive association between plaque bilirubin concentrations and NIRAF), and the observation that commercial bilirubin powder showed NIRAF at a broad range of excitation wavelengths [2]. Importantly, the latter evidence did not establish biological relevance. Indeed, the maximal excitation and fluorescence emission of the natural form of bilirubin in mammals, bilirubin IXα, is at 460 and 530 nm, respectively, whereas the NIRAF detected in unstable plaque used excitation and emission wavelengths of 730 and 803 nm, respectively. Moreover, bilirubin has an extremely weak fluorescence [11], independent of whether the pigment exists in its free or protein-bound form, and even when dissolved at higher-than-physiological concentrations. Together, these findings do not support the notion that bilirubin is the source of NIRAF in unstable plaque with hemorrhage, although they do not negate the potential utility of NIRAF as a non-invasive marker of intra-plaque hemorrhage, and hence plaque instability.

In addition to demonstrating that bilirubin is not the source of NIRAF in unstable atherosclerotic plaque, the NIRAF signal in Segment II being observed as higher than in Segment I indicates that elevated NIRAF is not a reliable indicator of the presence of IPH, or other characteristic features of unstable, high-risk plaque, such as cap thinning, inflammatory macrophages, and a necrotic core. Similar to Segment I, Segment II is characterized by outward remodeling. Unlike Segment I, however, Segment II contains a more cellular plaque that lacks other features of unstable plaque, including IPH, cap thinning, and the accumulation of lipid, inflammatory macrophages, and a necrotic core [7]. This raises the possibility of a false-positive identification of an unstable atherosclerotic plaque, based solely on an elevated NIRAF.

Consistently, our results indicate that an elevation of NIRAF in human coronary arteries may reflect the presence of a more complex, mature atherosclerotic disease. Thus, despite the recent interest in the potential of NIRAF to differentiate vulnerable from stable atheroma, it appears more plausible, based on our data, that NIRAF differentiates mature (AHA grade IV–VI) from immature plaques (AHA < III), rather than chiefly IPH or vulnerable and destabilized atheroma. A previous study, applying a dual modality optical coherence tomography (OCT) and NIRAF intravascular imaging system and catheter in human coronary arteries in vivo, reported that an elevated NIRAF was associated with a high-risk morphological phenotype, including OCT fibroatheroma, plaque rupture, and fibroatheroma associated with in-stent restenosis [3]. However, in that study, the NIRAF signal was limited, and its spatial distribution was more focal than OCT, with less than a quarter of the lipid OCT fibroatheromas, and less than half of the OCT macrophage accumulation, showing high NIRAF signals [3]. The authors concluded that NIRAF provides complementary information to that obtained by the structural imaging using OCT [3].

There are some limitations to this study. Firstly, we noted that the FLECT used by Htun et al. [2] was built by Bioscan, while the FLECT/CT used in this study was built by Trifoil InSyTe. We confirmed with Trifoil InSyTe that they had taken over Bioscan, and that the FLECT/CT system used in the present study had the same specification as Bioscan for in vivo NIRAF detection. Secondly, there is currently no in vivo NIRAF dataset to validate the current ex vivo NIRAF result, and further research is required, to establish the pre-clinical and clinical utility of in vivo NIRAF detection, before it can be a tool for identifying high-risk plaques.

## 4. Materials and Methods

### 4.1. Animal Study

*Bvra^−/−^Apoe^−/−^* mice on a C57BL/6J background, and corresponding littermate *Bvra^+/+^Apoe^−/−^* control animals [8], were provided with a Western diet (WD), and subjected to tandem stenosis (TS) surgery, as a model of plaque instability in bilirubin deficiency [7], with the modifications described previously [8,11]. Briefly, male *Bvra^−/−^Apoe^−/−^* and *Bvra^+/+^Apoe^−/−^* mice at six weeks of age were fed WD containing 22% fat and 0.15% cholesterol (SF00-219, Specialty Feeds, Glen Forrest, Australia) for 13 weeks. Six weeks after the commencement of the WD, TS surgery was performed to induce the formation of unstable plaques, with the operator blinded to genotype. The mice were anesthetized with 2–4% isoflurane, an incision was made in the neck, and the right common carotid artery was isolated from the connective tissues. Two ligatures were then placed on the carotid artery, with the distal ligature 1 mm from the right carotid artery bifurcation, and the proximal ligature 3 mm from the distal ligature. To control for, and hence to ensure, a consistent reduction in blood resulting from each ligature, blood flow through the right common carotid artery was measured before and after the placement of each ligature, using a perivascular flow module (TS420, Transonic, NY, USA) and a 0.7 mm perivascular flow probe (MA0.7PSB, Transonic, NY, USA). The ligatures were adjusted, such that the blood flow was reduced to 70% of the baseline after placement of the first (distal) ligature, and to 20% of the baseline after placement of the second (proximal) ligature. The incision was closed, and the animal then returned to the housing cages. All mice were housed in a temperature-controlled room, on a 12 h light/dark cycle, and were allowed access to water and food ad libitum for an additional seven weeks, after which they were euthanized, as described previously [7,8,12]. Segments I (unstable plaque), II (highly cellular plaque with features of stable plaque except for the presence of outward remodeling), IV (the lesion-free left carotid artery), and V (stable plaque) were collected (Figure 2 and references [8,9]). The fibrous cap thickness of the atherosclerotic plaque in the above arterial segments was used as the primary readout of plaque instability. The thickness of the cap was assessed histologically by the quantification of its collagen content, using picrosirius red staining visualized under polarized light. The intraplaque hemorrhage (assessed by the content of the red blood cells), hemosiderin, and fibrin were used as the secondary readouts to distinguish stable from unstable plaque.

All animal studies were approved by the Animal Welfare Committees of the Garvan Institute of Medical Research/St Vincent’s Hospital (Protocol: 16–33), and Sydney Local Health District (Protocol: 2020–027). All procedures were carried out according to the Guidelines for Animal Research outlined by the National Health and Medical Research Council of Australia, and the Guide for the Care and Use of Laboratory Animals outlined by National Institutes of Health.

### 4.2. Fluorescence Emission Computed Tomography (FLECT)

The in vivo measurement of the near-infrared autofluorescence (NIRAF) was performed using a FLECT/CT (Trifoil InSyTe, Chatsworth, CA, USA), seven weeks after the TS surgery, as described previously [9]. Briefly, the mice were anesthetized and the fur removed as required, and the animals then placed in the imaging chamber, where they remained anesthetized throughout the imaging procedure (~70 min). The X-ray CT scans were performed using the following settings: 30 kV for tube voltage, 500 µA for tube current, and 150 ms exposure time. The fluorescence scans were performed using a 730 nm excitation laser and 803 nm filter in step-and-shoot scanning mode, with 29 source angles per slice and 500 ms exposures, and subsequent fluorescence attenuation scans were performed in continuous mode. A reconstructed 3D image of the x-ray scan detailing the anatomy of the mouse was overplayed onto the corresponding fluorescence-reconstructed image by VivoQuant 3.5 (InviCRO, Needham, MA, USA).

### 4.3. Determination of Ex Vivo Near-Infrared Autofluorescence (NIRAF) in Mouse Tissue

The ex vivo measurement of NIRAF was performed using an Odyssey infrared imaging system (CLx, Li-COR, Lincoln, NE, USA), and different arterial segments isolated from TS mice. Figure 2 shows a representative image of arterial segments free of atherosclerotic disease, and containing plaques of different phenotypes: Segments I (unstable plaque), II (highly cellular plaque with features of stable plaque except for the presence of outward remodeling), IV (the lesion-free left carotid artery), and V (stable plaque) were collected and placed into the imaging system. A serially diluted fluorescence dye (IRDye800CW, LI-COR, Lincoln, NE, USA) was used as a reference for quantitative purposes. The fluorescence intensity scans were performed using the following settings: 800 nm channel (excitation = 785 nm, emission > 800 nm), intensity level at 5, resolution at 21 μm, image quality at medium. Image Studio (LI-COR, Lincoln, NE, USA) was used to quantify the fluorescence intensity.

### 4.4. Human Study

The clinical study was approved by the Human Research Ethics Committee of St Vincent’s Hospital, Darlinghurst, Australia (Approval Number: 2022/PID02661), and conformed to the ethical guidelines of the 1975 Declaration of Helsinki. All participants provided written informed consent. The coronary arteries from ten native hearts of transplant recipients with established ischemic cardiomyopathy were procured. Within an hour of the cardiac explant, the coronary trees were dissected from the myocardium. The arteries were located by examining the aortic root for the left and right coronary cusps, and finding the coronary ostia. To facilitate dissection along the correct plane, a histology probe was used to gently engage the coronary ostia. The arteries were removed from the myocardium from the proximal to distal segments, using a scalpel and surgical blade, blunt dissection, and scissors along the underside of the epicardial vessels, preserving the proximal portion of the side branches of the main vessels.

After the removal of the coronary trees and the surrounding adipose tissue, the arteries underwent a previously described ex vivo coronary imaging technique for plaque characterization [10]. Thirty plaques were categorized as calcific or non-calcific using this method, before they underwent NIRAF signal intensity determination, and total cholesterol concentration, as described below. In selected specimens, calcific and non-calcific plaques were then histologically examined, using the AHA consensus group’s histological classification of atherosclerosis [13,14], to confirm the presence of mature lesions (AHA IV–VI) and describe their composition.

### 4.5. Determination of the NIRAF of Human Atherosclerotic Plaque

For human samples, the ex vivo NIRAF of the atherosclerotic plaque was determined using a Bio-Rad ChemiDoc™ MP Imaging System (Hercules, CA, USA), utilizing the preconfigured IRDye 800CW Blot setting (excitation = 755–777 nm, emission > 800 nm) and an exposure time of 5 s. The surrounding fat was removed from the vessel, and arterial tissues containing plaques were placed in 1.5 mL Eppendorf tubes, and imaged along with a positive control with serial dilutions of IRDye 800CW (LI-COR, Lincoln, NE, USA). For the quantification of NIRAF intensity, Bio-Rad Image Lab software (version 6.0.1) was used to trace around each coronary specimen, and the mean intensity was recorded.

### 4.6. Measurement of Cholesterol Using HPLC

Following the determination of NIRAF, de-identified arterial samples were homogenized and saponified, as described previously, for subsequent cholesterol determination [15]. Briefly, the samples were weighed prior to homogenization in 50 µL PBS containing 50 µM diethylenetriamine penta-acetic acid, and methanol (450 µL) was used as an extraction buffer. The samples were then saponified by heating for 60 min at 80 °C, after the addition of 50 µL 10 N sodium hydroxide. After cooling to room temperature, 200 µL water was added to each sample, before the addition of 500 µL n-hexane, rigorous mixing, and centrifugation of the sample at room temperature for 5 min at 500 RCF. The upper hexane layer was removed, and transferred to HPLC vials. The samples were dried with nitrogen, and the n-hexane extraction was repeated, before the dried lipids were reconstituted in ethanol for the determination of cholesterol using HPLC. The cholesterol was quantified using an Agilent 1100 Series HPLC system, equipped with a Supelcosil LC-18 column (25 cm × 4.6 mm, 5 µm), C18 guard column, and a diode array detector set at 214 nm. For each sample, 100 µL ethanolic extract was injected onto the column, and the column eluted at 1 mL/min for 30 min, using ethanol/methanol/isopropyl alcohol/1 M ammonium acetate pH 4.4 (65/30/3/2, vol/vol/vol/vol) as the mobile phase. Under these conditions, the cholesterol eluted at 7.9 min, and the peak area was integrated using Agilent ChemStation software (version B.04.03). Each sample was analyzed in triplicate, and the average cholesterol content per sample was then calculated. A standard curve, prepared from injections of known amounts of authentic cholesterol dissolved in isopropyl alcohol, was used to quantify the cholesterol content in each sample by area comparison, with the obtained results normalized to tissue wet weight.

### 4.7. Statistical Analyses

Analyses were performed using Graphpad Prism 9. Results were expressed as mean values, with error bars representing the standard error of the mean (SEM). Descriptive statistics were used to present variables, their mean values, and standard deviations. The data were first assessed for normality using the D’Agostino–Pearson test. For normally distributed data, univariate parametric statistics were performed using Student’s *t*-tests for pairwise comparisons. For non-normally distributed data, Mann–Whitney U tests were performed. Linear regression analyses were used to correlate the cholesterol concentrations with the NIRAF. The statistical significance was defined at a two-tailed *p*-value of <0.05.

## 5. Conclusions

In conclusion, by demonstrating the dissociation between NIRAF and the presence of bilirubin and IPH, the present study establishes that IPH and its associated catabolism of heme to bilirubin are not reliable molecular signatures of NIRAF. In the context of previous studies, our results indicate that the compositional meaning of an elevated NIRAF in atherosclerotic plaque remains obscure. It is important to note that the NIRAF signal in unstable plaques is likely multifactorial, and the contributions from different sources can vary depending on the specific characteristics of the plaque and the stage of atherosclerosis. Further research is needed to elucidate the relative contributions of these sources, and to identify additional factors that may contribute to NIRAF in unstable plaques.

## Figures and Tables

**Figure 1 ijms-24-10727-f001:**
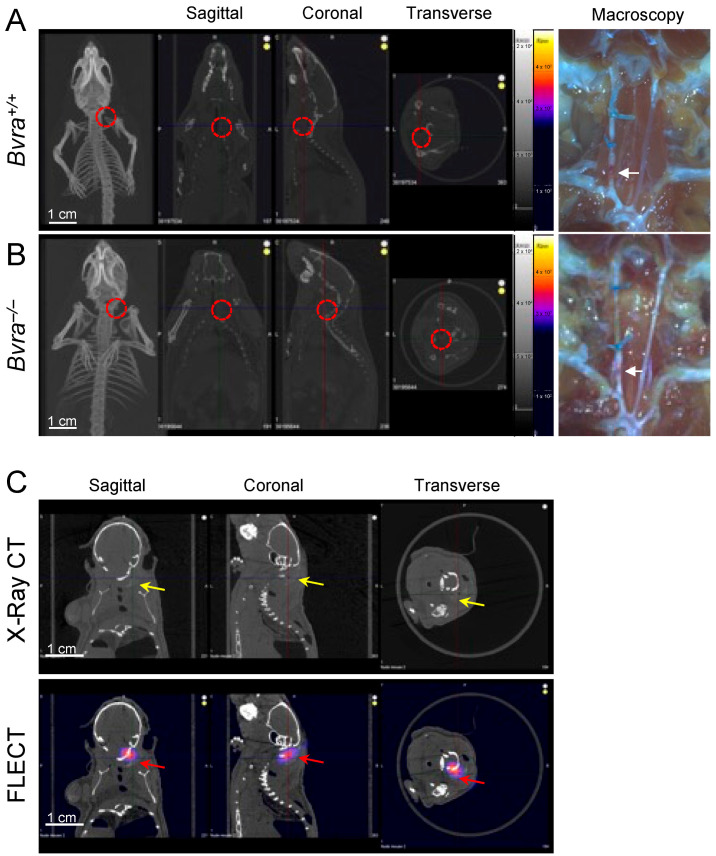
The determination of in vivo near-infrared autofluorescence (NIRAF). (**A**,**B**) Tandem stenosis (TS) surgery was performed on *Bvra^+/+^Apoe*^−/−^ and *Bvra^−/−^Apoe*^−/−^ mice. Shown are representative reconstructed in vivo fluorescence emission computer tomography (FLECT/CT) images, showing the absence of NIRAF in atherosclerotic plaque in the carotid arteries of *Bvra^+/+^Apoe*^−/−^ and *Bvra^−/−^Apoe*^−/−^ TS mice (the open circle indicates the location of the right carotid artery). The right panels show the corresponding macroscopic images depicting unstable plaque with IPH (white arrows). (**C**) The X-ray contrast reagent Fenestra™ (1:5 vol/vol), and fluoroprobe Cy7 (5 ng/L), were diluted and mixed in 2% agarose gel. This mixture (100 µL) was injected at the location of the right carotid artery in a naïve mouse carcass. The representative reconstructed FLECT/CT shows the obvious x-ray signal of Fenestra (yellow arrows) and the NIRAF signal of Cy7 (red arrows) in the carcass. A 730 nm excitation laser and 803 nm filter were used, and the scale bar = 1 cm. Part of Panel A was modified from reference [9].

**Figure 2 ijms-24-10727-f002:**
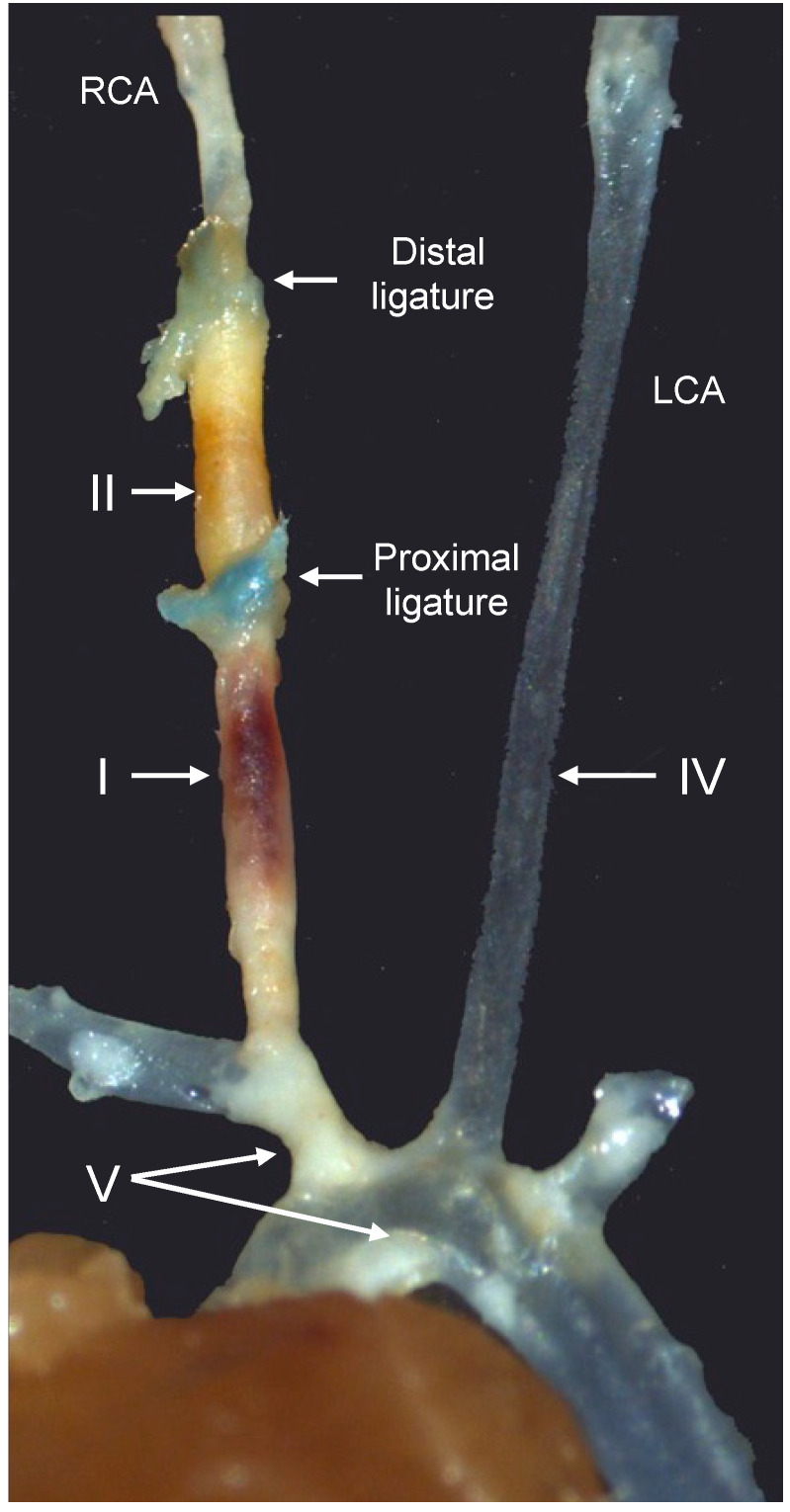
Representative image of the different arterial segments in the tandem stenosis (TS) mouse model of plaque instability, seven weeks after the placement of the distal and proximal ligatures. Arterial segments I, II, IV, and V represent unstable plaque with visible intraplaque hemorrhage, highly cellular plaque with features of stable plaque except for the presence of outward remodeling, the lesion-free left carotid artery (LCA), and stable plaque in the brachiocephalic trunk and aortic arch, respectively. RCA, right carotid artery. Modified from reference [9].

**Figure 3 ijms-24-10727-f003:**
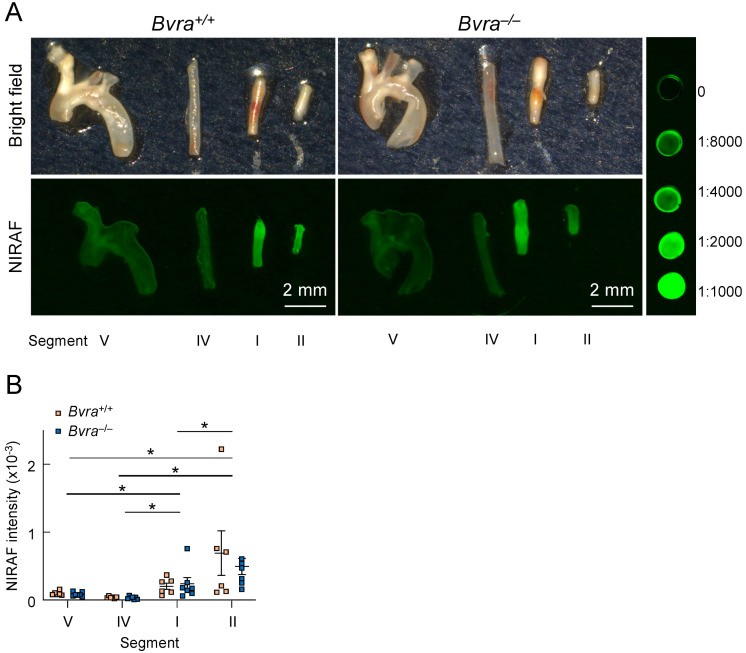
Near-infrared autofluorescence (NIRAF) in the arteries of *Apoe*^−/−^ tandem stenosis (TS) mice. (**A**) Representative ex vivo bright-field and Odyssey (CLx, Li-COR, Lincoln, NE, USA) NIRAF images of Segments I (unstable plaque), II (highly cellular plaque with features of stable plaque except for the presence of outward remodeling), IV (the lesion-free left carotid artery) and V (stable plaque) from male *Bvra*^+/+^*Apoe*^−/−^ (top two panels) and *Bvra*^−/−^*Apoe*^−/−^ TS mice (bottom two panels), seven weeks after TS surgery, in comparison with in comparison with serial dilution of IRDye800CW (LI-COR, Lincoln, NE, USA). The fluorescence intensity scans were performed using the 800 nm channel (Ex_785 nm_, Em_>800 nm_), intensity level 5, resolution at 21 μm, and image quality set at medium, and the fluorescence intensity was quantified using Image Studio (LI-COR, Lincoln, NE, USA). Scale bar, 2 mm. (**B**) The quantification of the NIRAF intensities of Segments I, II, IV, and V of six *Bvra^+/+^Apoe^−/−^* and seven *Bvra^−/−^Apoe^−/−^* TS mice, as determined by Odyssey NIRAF. Individual data and mean ± SEM are shown, with * indicating significant (*p* < 0.05) difference between the different segments (Mann–Whitney). There were no significant differences between the *Bvra^+/+^Apoe^−/−^* and *Bvra^−/−^Apoe^−/−^* TS mice for any of the four segments analyzed. Modified from reference [9].

**Figure 4 ijms-24-10727-f004:**
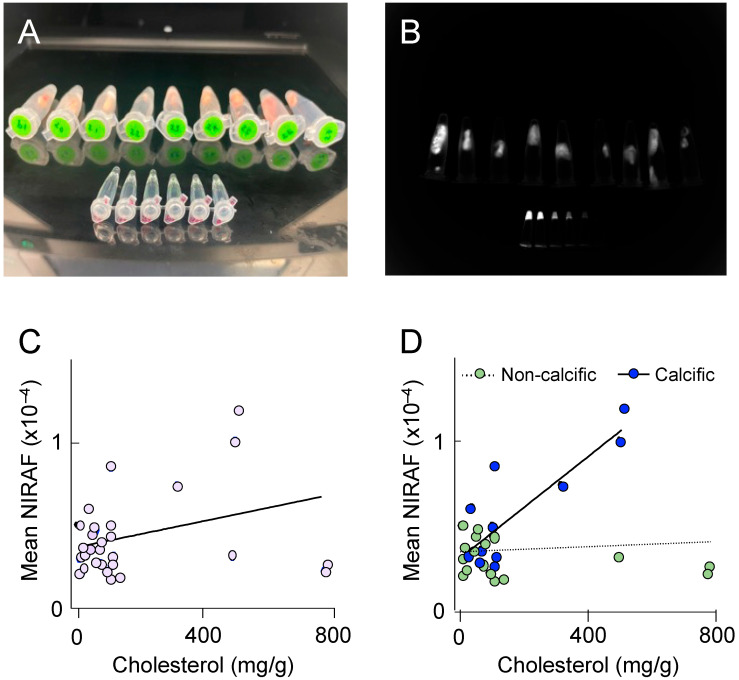
Near-infrared autofluorescence (NIRAF) in human coronary arteries. (**A**) Photograph of individual coronary plaques placed in 1.5 mL Eppendorf tubes, with the smaller Eppendorf tubes in the foreground containing serial dilutions of IRDye800CW (LI-COR, Lincoln, NE, USA) shown from left to right. (**B**) Representative image of the NIRAF quantification. Each plaque was outlined, and its mean NIRAF determined. (**C**,**D**) The correlation between the total cholesterol concentration determined using high-pressure liquid chromatography with UV_210 nm_ detection, and the NIRAF determined using Odyssey ex vivo, in 30 human coronary plaques. (**D**) The NIRAF signal intensity correlated positively with the cholesterol in calcific plaques (*n* = 11, R^2^ = 0.69, *p* = 0.001), but not in non-calcific plaques (*n* = 19, R^2^ = 0.01, *p* = 0.6), when determined using ex vivo computed tomography.

## Data Availability

The data presented in this study are available on request from the corresponding author.

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
