# Peer review of "Near-Infrared Autofluorescence (NIRAF) in Atherosclerotic Plaque Dissociates from Intraplaque Hemorrhage and Bilirubin"

_ijms, 2023, doi:10.3390/ijms241310727_

Round 1
Reviewer 1 Report
This is a novel and interesting paper that uses near infrared autofluorescence (NIRAF) technology in atherosclerotic plaques to reveal unstable plaque lesions in vivo and ex vivo in a mouse model. However, there are a few problems with this paper.
Specific comments
We believe that the scientific basis for determining whether a plaque is unstable or not is not well described. The authors should provide more details on this point.
In Fig2, an in vivo FLECT image is shown, can authors get a clearer image? Also, is it possible to present an image that can be contrasted with the NIRAF image in Fig3A? If possible, please try to make the location of the artery in the in vivo FLECT image in Fig 2 a little more understandable to all readers.
Authors show a positive correlation between cholesterol content and NIRAF in calcified lesions in the vessel wall, however it doesn't calcification itself emit an autofluorescent signal that increases NIRAF?
In the introduction, there is a very poor explanation of the state of research to date on plaque imaging and NIRAF.
Author Response
This is a novel and interesting paper that uses near infrared autofluorescence (NIRAF) technology in atherosclerotic plaques to reveal unstable plaque lesions in vivo and ex vivo in a mouse model. However, there are a few problems with this paper.
Response: We thank the Reviewer for the overall positive assessment of our work, as well as the detailed comments.
Specific comments
We believe that the scientific basis for determining whether a plaque is unstable or not is not well described. The authors should provide more details on this point.
Response: We thank the Reviewer for this constructive suggestion. In the mouse model of plaque instability, stable and unstable plaques were distinguished using the criteria provided by the original paper describing this model (Circ Res2013;113:252). Briefly, fibrous cap thickness (assessed by quantification of the collagen content after picrosirius red staining and visualization under polarized light) was used as primary readout of plaque instability. Intraplaque hemorrhage (assessed by the content of red blood cells), hemosiderin and fibrin in the plaque were used as the secondary readouts to distinguish stable from unstable plaque. This information is now provided on page 8, lines 248-253 of the revised manuscript.
For human coronary atherosclerotic plaque, ex vivo CT was used as a robust imaging tool to determine the presence of calcification. Once plaques were confirmed to be calcific or
non-calcific, a select number underwent confirmatory histological assessment using the American Heart Association’s (AHA) consensus group’s histological classification of atherosclerosis (Arterioscler Thromb Vasc Biol2000;20:1177; ibid 2000;20:1262 –included as new references 13 & 14 in the revised manuscript) whereby the presence of mature (AHA IV-VI) atheroma was confirmed and plaque composition was described. We have added this information on page 9, lines 301-304 of the revised manuscript.
In Fig2, an in vivo FLECT image is shown, can authors get a clearer image? Also, is it possible to present an image that can be contrasted with the NIRAF image in Fig3A? If possible, please try to make the location of the artery in the in vivo FLECT image in Fig 2 a little more understandable to all readers.
Response: In the revised manuscript, we now provide a clearer image with high resolution for Figure 2. The FLECT image of a Bvra+/+ TS mouse shown in Figure 2 corresponds to the NIRAF image of the Bvra+/+ TS mouse shown in Figure 3A. Also, we have added open circles to indicate the location of the right carotid artery in the in vivo FLECT images in Figure 2, to make these images more readily understandable to the readers.
Authors show a positive correlation between cholesterol content and NIRAF in calcified lesions in the vessel wall, however it doesn't calcification itself emit an autofluorescent signal that increases NIRAF?
Response: Calcification and NIRAF can be observed in some similar pathological conditions, e.g., atherosclerosis, although to the best of our knowledge, they are not directly associated with each other. Indeed, intracoronary NIRAF assessment has demonstrated that calcific lesions do not emit NIRAF signal (JACC Cardiovasc Imaging 2016;9:1304). As such it would appear from our findings and those of others that neither calcification or the lipid pool or hemorrhage are solely responsible for NIRAF signal, but rather mature and complex lesions appear to contain higher signal in human studies. The specific source of NIRAF in such lesions remains to be elucidated.
In the introduction, there is a very poor explanation of the state of research to date on plaque imaging and NIRAF.
Response: In the revised manuscript, we now provide additional information on the current state of research on plaque imaging and NIRAF (page 2; lines 48-53), and we discuss plaque imaging and NIRAF on page 7, lines 219-222.
Reviewer 2 Report
The authors present an original study entitled “Near infrared autofluorescence (NIRAF) in atherosclerotic plaque dissociates from intraplaque hemorrhage and bilirubin”
The results of the presented study are interesting, novel, and have clinical significance. The article has a good scientific soundness. The authors used an original approach that ensures the validity of the results obtained. The use of two lines of mice, one of which had the Biliverdin reductase a gene knocked out, provided convincing evidence that bilirubin has no effect on NIRAF.
I would recommend adding a brief description of the study's limitations to the Discussion. In addition, it encourages the authors to add possible explanations for the NIRAF in mature plaques, and outline directions for future research.
Author Response
The authors present an original study entitled “Near infrared autofluorescence (NIRAF) in atherosclerotic plaque dissociates from intraplaque hemorrhage and bilirubin”
The results of the presented study are interesting, novel, and have clinical significance. The article has a good scientific soundness. The authors used an original approach that ensures the validity of the results obtained. The use of two lines of mice, one of which had the Biliverdin reductase a gene knocked out, provided convincing evidence that bilirubin has no effect on NIRAF.
Response: We thank the Reviewer for the overall positive assessment of our work, as well as the detailed comments.
I would recommend adding a brief description of the study's limitations to the Discussion. In addition, it encourages the authors to add possible explanations for the NIRAF in mature plaques, and outline directions for future research.
Response: We thank the Reviewer for this valid suggestion. We now have added a brief description of the study's limitations to the Discussion of the revised manuscript (page 7, lines 215-222). We provide possible explanations for the NIRAF signal in mature human plaques (page 7, lines 201-214), as well as additional information on the current state of research in imaging and NIRAF (page 2, lines 48-53) and directions for future research (and page 10, lines 350-354).
Reviewer 3 Report
The authors investigated NIRAF of atherosclerotic plaques in mice models as well as human coronary artery samples. However, I have some comments.
1) In Methods, the authors should describe and summarize the methods of statistical analysis.
2) It seems to be difficult to read this paper. A thorough review of this paper should be performed by a native English speaker.
It seems to be difficult to read this paper. A thorough review of this paper should be performed by a native English speaker.
Author Response
Comments and Suggestions for Authors:
The authors investigated NIRAF of atherosclerotic plaques in mice models as well as human coronary artery samples. However, I have some comments.
1) In Methods, the authors should describe and summarize the methods of statistical analysis.
Response: We have now added statistical analysis to the method section of the revised manuscript.
2) It seems to be difficult to read this paper. A thorough review of this paper should be performed by a native English speaker.
Response: A thorough review of this manuscript has been performed by a native English speaker.
Round 2
Reviewer 1 Report
no further comments